# Family Functioning and Suicide Attempts in Mexican Adolescents

**DOI:** 10.3390/bs13020120

**Published:** 2023-02-01

**Authors:** Francisco Alejandro Ortiz-Sánchez, Aniel Jessica Leticia Brambila-Tapia, Luis Shigeo Cárdenas-Fujita, Christian Gabriel Toledo-Lozano, María Alejandra Samudio-Cruz, Benjamín Gómez-Díaz, Silvia García, Martha Eunice Rodríguez-Arellano, Edgar Oswaldo Zamora-González, Luz Berenice López-Hernández

**Affiliations:** 1Life Cycle Department, Autonomous University of Guadalajara, Zapopan 45129, Mexico; 2Mental Health Department, Saint John of God Psychiatric Hospital, Zapopan 45100, Mexico; 3Basic Psychology Department, University Center of Health Sciences (CUCS), University of Guadalajara, Jalisco 44340, Mexico; 4Clinical Research and Genomic Medicine, Institute of Security and Social Services for State Workers, México City 03100, Mexico; 5Neuroscience Division, National Institute of Rehabilitation, México City 14389, Mexico; 6Genomic Medicine Service, National Institute of Rehabilitation, México City 14389, Mexico; 7North University Center, University of Guadalajara, Jalisco 46200, Mexico

**Keywords:** suicide attempt, adolescents, family, cohesion, FACES III

## Abstract

Suicide is considered a public health problem that affects families worldwide. Family functioning is the capability of the family system to fulfill needs during the stages of its development. In this study, we focused on evaluating family cohesion and adaptability in a group of adolescents who had attempted suicide and were hospitalized at a hospital for mental health disorders, compared to a control group. Methods: based on Olson’s circumplex model, we used the FACES III scale to gain insights into the family functioning of both suicidal and control groups. Results: The case group presented lower scores in cohesion and adaptability compared to the control group, with moderate effect-size for cohesion (Cohen’s d/r test = 1.217/0.52) and low effect-size for adaptability (Cohen’s d/r test = 0.746/0.35) (*p* < 0.001 for both variables), and also presented predominantly disengaged families (72.5% in the case group vs. 27.5% in the control group) and structured families (45% in the case group vs. 23.8% in the control group). The type of family described by the adolescents with a history of suicide attempts may explain the presence of low self-esteem and little emotional support usually present in this type of patient.

## 1. Introduction

Suicide is a complex phenomenon caused by interactions of biological, individual, family and social factors [1]. In Mexico, mortality statistics report 7818 deaths due to self-inflicted injuries in 2020, which represents a rate of suicide of 6.2 per 100,000 inhabitants, higher than that registered in 2019, of 5.65 [2]. Interestingly, studies regarding suicide attempts in teenagers are scarce in Mexico; nonetheless, a recent report demonstrated that suicide in children from 10 to 14 years old increased by 37%, and those of female adolescents aged 15 to 19, increased by 12% [2]. Similarly, between 2018 and 2020, suicidal ideation in adolescents increased from 5.1% to 6.9%, and suicidal behavior from 3.9% to 6% [2]. In addition, it has been documented that up to 51% of suicide attempts are due to family problems, in Mexico [3]. 

Different authors have proposed conceptual models in which family interactions, alterations, or situations of conflict precede the onset of the illness, where the family as a small group is part of the social context that influences the development and maintenance of the illness; among these authors, Minuchin and collaborators (1975) [4] studied the family process that fosters somatization, based on a set of etiological factors such as physiological vulnerability, the role of the unwell child in avoiding family conflicts, and four transactional characteristics that are observed in the behavior of a family, such as overprotection, lack of conflict resolution, rigidity and enmeshment [4]. 

Similarly, it has been reported that when family functioning is negative or dysfunctional, it is an agent that can lead to vulnerability to disease [5]. In this regard, risk factors have been identified that make families vulnerable and lead to the appearance of self-destructive behaviors among their members. Negative feelings and emotions that lead to hostile coexistence, lack of affectivity, and resentment, are characteristics that predominate in families of individuals with suicide attempts [6]. 

Among the family risk-factors associated with suicidal behavior are disagreement with norms, leading to family disorganization, lack of clarity in roles, poor and confrontational communication, hostility and feelings of rejection (the latter related with dysfunctional family cognitions), constant conflicts due to disagreements, transgression of limits, physical or psychological aggression, abandonment (emotional or physical) of one or both parents, as well as feelings of hopelessness and pessimism, among others [6,7,8]. Moreover, role saturation, power disputes between members, inflexibility in problem-solving, low tolerance, and power conflicts that destabilize family cohesion, are other factors that, if present with those already mentioned, can predispose someone to suicidal behavior [6].

In a systematic review conducted in 2018, it was identified that the family variables most associated with suicidal ideation and attempts in Mexico were: family problems, detachment from the family scenario and presenting negative emotions associated with the family context [9]; however, the characteristics of family functioning were not clearly described. Therefore, the analysis of family functioning is of constant interest [10], due to its contribution to family processes. 

Family functioning is the capability of the family system to fulfill needs during the stages of its development, fundamentally in the affective aspects, socialization, care, reproduction and family status, in accordance with the norms of the society to which it belongs. Family functioning allows for the facing and overcoming of the crises which the family goes through [11,12]. Although a previous study showed a negative correlation between family cohesion and flexibility and problem-focused coping with suicidal ideation in adolescents [13], other reports showed that lower family cohesion and hopelessness were associated with suicidal ideation in adolescents [14,15,16]; however, none of those studies were performed in adolescents hospitalized for suicidal attempts. On the other hand, some studies have shown that in families with low cohesion and adaptability and, therefore, poor systemic functioning, suicidal ideation and suicide attempts are more frequent [17]. A study on adolescent suicide attempts reported lower cohabitation, mutual support, and performance of joint activities with the family, which were indicators of family disunity [3]. Other studies on the assessment of family functioning point out that coping is one of the dimensions that correlate with suicidal ideation, and is a variable that can predict it [18].

In general, in terms of cohesion and adaptability, it has been observed that families of suicidal individuals are characterized by a lack of physical and emotional unity in the face of everyday situations, intolerance, and hostility that hinder their cohesion, i.e., their rapport, commitment, and mutual support; in addition, these families are characterized by the inability to adapt to present demands by performing internal changes according to the circumstances, due to inflexibility in modifying their functions, their rules (implicit or explicit) and their power structures [6,17,19].

In contrast, the relevance of both family cohesion and adaptability in enabling young people to contend with stress has been noted, and family cohesion has been described as a protective factor for suicide, reducing suicidal thoughts and risk behaviors [17,20]; therefore, among the central factors on which some therapies for the care of families with suicidal members are focused, is family cohesion, with which they manage to reduce suicidal ideation in individuals with depression [21].

One of the most relevant models for understanding family systems is the circumplex model developed by Olson (1986), which proposes that cohesion (defined as the degree of emotional union perceived by family members), adaptability (considered as the magnitude of change in roles, rules and leadership experienced by the family) and communication are the three dimensions that mainly define family functioning [22]. Olson (1986) considers that the interaction of cohesion- and adaptability-dimensions condition family functioning [23]. Each of these dimensions has four traits, and their combination determines 16 family types, integrated into three levels of family functioning (extreme, mid-range and balanced) (Figure 1).

Based on this model, the scale of Assessment of Adaptability and Family Cohesion (FACES) has been generated [24,25], and this allows the identification of the level of adaptability and cohesion of a family system, as well as classifying it into eight sub-types (four according to cohesion scores and four according to adaptability scores,) and finally into three types: extreme, mid-range or balanced (Olson, 1991). Balanced family systems tend to be more functional, while extremes are more problematic throughout the life cycle [22].

In this study, we focused on evaluating family cohesion and adaptability from the point of view of Olson’s circumplex model, through the FACES III scale, in a group of adolescents who attempted suicide and were ministered to within a hospital for mental health disorders, compared with a control group (with no suicide attempts). In this sense, the main hypothesis of the present study was that the group of cases would present a different distribution of family types according to cohesion and adaptability, compared to the control group. Nevertheless, additional hypotheses were also explored as follows: (1) the group of cases would differ in sociodemographic variables, with respect to the control group, (2) the group of cases has less family cohesion and family adaptability than the control group, and these are therefore protective factors for suicide attempts in adolescents hospitalized for this reason, and finally, (3) the third hypothesis is that the types of families proposed by Olson’s circumplex model (extreme, mid-range or balanced) are also significantly different among groups, finding fewer balanced families in the group of cases. 

**Figure 1 behavsci-13-00120-f001:**
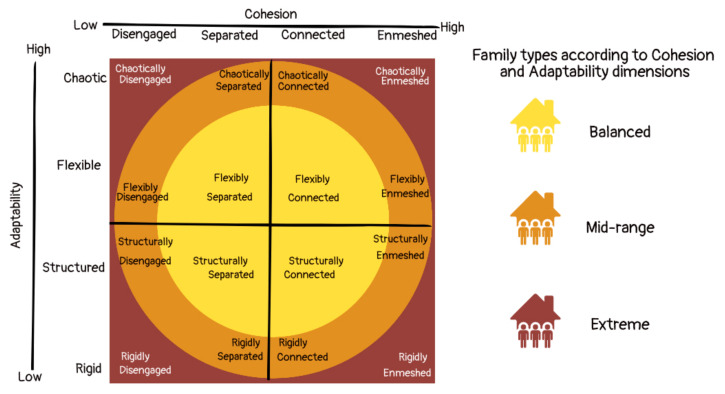
Diagram of Olson’s circumplex model showing the family types (modified from Hanson and Keplinger, 2020) [26].

## 2. Subjects and Methods

To carry out this research, a case-control study was performed, and permission was requested from the San Juan de Dios Hospital, located in Zapopan, Jalisco, Mexico. From 1 November 2019 to 1 February 2020, all the adolescents aged 12–17 who were hospitalized in the aforementioned institution were invited to participate, with the main reason for hospitalization being a suicide attempt, regardless of the method used or their adjacent diagnosis. A hospital stay of at least one day was considered as an inclusion criterion, to ensure that they were more stable and able to participate in the study. After explaining the objectives of the project, informed consent was obtained from the adolescents and their parents.

For the recruitment of the control group, secondary schools and a high school in the area were visited. We looked for two matching controls by age and sex for each case. The students were invited to participate voluntarily, and they were given an informed consent form to be reviewed and signed by at least one of the parents. At that time, the content of the FACES III instrument and how to fill in the datasheet were explained to participants.

Once all the data was collected, an SPSS database was generated for statistical analyses.

### Methods

All participants completed a questionnaire with the following socio-demographic variables: age, sex, schooling, number of siblings, school grades, maximum parental schooling, parental marital status (measured as married, divorced, single mother, single father, separated, free union and widow) and housing conditions (own, rented and borrowed house) (Table 1).

The psychiatric diagnosis and the number of suicide attempts were also recorded. 

The FACES III instrument [23,27] is the third version of the FACES series of scales, and has been developed to assess two of the main dimensions of the circumplex model: family cohesion and adaptability. FACES III in Mexican Spanish has been validated by Gómez-Clavelina in 1999, who reported that the instrument is reliable (Cronbach’s alpha > 0.70) and showed an adequate factorial structure [25,28]. The FACES III scale contains 20 items: the 10 odd ones evaluate family cohesion and the 10 even ones, family adaptability [29]. The items are scored using a 5-point Likert-type scale, ranging from 1 = almost never, to 5 = almost always. Higher scores on the scales of cohesion and adaptability tend to reflect more balanced family functioning, whereas lower scores indicate more unbalanced family functioning. However, very low or very high levels in these two variables are related to extreme-type families.

**Statistical analysis:** To describe qualitative variables, we used frequencies and percentages and for quantitative ones, means and standard deviation. To compare qualitative variables between both groups we used the chi-squared test. In order to compare the psychological and sociodemographic quantitative variables between groups, we used Student’s *t*-test and the Mann–Whitney U test for parametric and non-parametric distributions, respectively. To perform comparisons between the psychological variables, we used the Spearman correlation test, considering the non-parametric distribution of the variables. In order to detect the distribution of the data, the Kolmogorov–Smirnov test was used. All statistical analyses were carried out using the software SPSS v. 25.0 and a *p* value of <0.05 was considered as statistically significant. To calculate effect sizes (Cohen’s d and Cohen’s r tests) for the variables of cohesion and adaptability, we used the JASP software.

## 3. Results

In total, 120 participants were included, with 40 of them being cases used in the study: 32 women (80%) and 8 men (20%). The 80 adolescents (matched by age and sex) were included in the control group of 60 women (75%) and 20 men (25%). All adolescents reported being single, and were residents of the metropolitan area of the city of Guadalajara. The socio-demographic characteristics of the participants are described in Table 1.

With regard to socio-demographic variables, when the variable parent’s situation is compared with the rest of the parental marital-situations, it was shown that having married parents almost reaches statistical significance as a protective factor, with an odds ratio = 0.65, and a confidence interval ([0.44–0.96], *p* = 0.05). Interestingly, another factor according to this study is having >85 points in school grades, since it has a protective effect against suicidal attempts, although it can also be a consequence of not having psychological problems that could lead to suicidal attempts, with an odds ratio = 0.571 CI (0.36–0.89, *p* = 0.027). 

In the comparison of the two subscales of the instrument FACES III, cohesion and adaptability, we found that patients had significantly lower levels of both variables, being even lower for cohesion, which is corroborated with the effect size of the differences found, which is moderate for cohesion (Cohen’s r test = 0.52) and low for adaptability (Cohen’s r test = 0.35) (Table 2). These observations were corroborated when we correlated the levels of these variables with being a case, where the correlation with cohesion was r = −0.491, *p* < 0.001, and for adaptability was r = −0.359, *p* < 0.001.

When the number of suicidal attempts in the group of cases was tested against cohesion and adaptability scores, we observed negative although not significant correlations: suicidal attempts with adaptability: r = −0.201, *p* =0.214 and suicidal attempts with cohesion: r = −0.193, *p* = 0.232. 

With respect to the family sub-types according to the scores of cohesion and adaptability, statistically significant differences were found between groups in both variables (*p* < 0.001). In cohesion, in the case group there was a predominance of disengaged families when compared with the control group: 72.5% vs. 27.5%. Additionally, there were more structured families in the case group when compared with the control group (45.0% vs. 23.8%) and fewer chaotic families when compared with the control group (12.5% vs. 43.7%) (Table 3). 

With regard to family type, for the frequency of families belonging to the categories extreme, mid-range and balanced families, the distribution was not different (Table 4). However, we found a higher frequency of balanced families in the control group, and we observed a different distribution in the typology of Olson’s circumplex model (Figure 2 and Appendix A).

## 4. Discussion

The present study aimed to evaluate family cohesion and adaptability from the point of view of Olson’s circumplex model through the FACES III scale, in a group of adolescents who attempted suicide and were ministered to in a hospital for mental health disorders, compared with a matched control group.

With respect to our first hypothesis, our data showed significant differences among groups with regard to academic performance, which could be a consequence of the mood of the patients with a history of suicide attempts or an additional risk factor for the negative perception and low self-esteem that is usually described in these patients. It has been suggested that adolescents with low academic performance present more frequent risk behaviors (substance use, early sexual relations, suicide attempts and antisocial behavior) than young people with high school-performance [30]. On the other hand, González-Forteza et al. (2002) [31] described the fact that among the reasons why adolescents execute suicide attempts, the most frequent precipitating event was poor school-performance.

Regarding the comparisons of sociodemographic variables associated with parental characteristics (schooling, marital status) and housing, it was only identified that having married parents proved to be a protective factor against suicide attempts. The study conducted by Park and Park [32] identified the fact that there is a higher risk of suicidal behavior in adolescents living without their parents or in reconstructed families. On the other hand, high rates of divorce and separation in the parents of adolescents with a history of suicide attempts have been identified [25]. This background allows us to consider that the presence of family systems in which high instability and detachment, and high levels of stress and drastic changes occur (predominantly in divorce, separation and reconstitution of the families, without excluding even marriage) generate vulnerability to mental illness, largely due to the poor quality of the relationship among their members [33]. 

In line with our other hypotheses, we observed that cases showed significantly lower levels of both variables (cohesion and adaptability), which, additionally, were also negatively (although not significantly) correlated with the number of suicide attempts in the group of cases. This denotes the importance of promoting these two characteristics in family functioning in order to prevent suicide attempts in the Mexican population. In addition, we also confirmed that the sub-types of the two big dimensions of cohesion and adaptability (e.g., disengaged, rigid, connected or structured) were significantly different between both groups, with the highest difference found in the sub-type of disengaged families, being significantly higher in the group of cases. Finally, in line with our last hypothesis, we did not find significant differences between the types of families proposed by Olson’s circumplex model, in this sense; although we found more balanced families in the control group, this difference was not statistically significant. 

Although there were no differences among the groups in the type of family (extreme, mid-range or balanced), the distribution in each group was different; in the case of adolescents with a history of suicide attempts, 60% were in the mid-range, while the controls, in which a greater frequency of balanced families, which tend to be more functional than the rest, would be expected, they were similarly distributed between the balanced (27.5) and extreme categories (23.8%), with the highest percentage in the mid-range (48.7%); however, a higher frequency of balanced families was found in the control group when compared with the group of cases (27.5% vs. 15%), although this difference did not reach statistical significance. The distribution of the family type in the control group in this study is relatively similar to that found in a sample of families with patients with non-psychiatric illnesses assigned to a family medicine unit in Veracruz, Mexico [34], in which 43% were identified as middle-range, 41% as balanced and 16% as extreme families. In this study, although we did find a lower proportion of balanced families in the control group when compared with this previous report, this proportion was higher than that found in the group of cases. There are no conclusive data on the types of families present in Mexico and whether they correspond to the ideal proposed by Olson. 

With regard to the features of cohesion and adaptability, it was found that the group of adolescents with a history of suicide attempts presented predominantly disengaged families (72.5%), which is significantly different to the proportion of these families in the control group (27.5%), being disengaged families the category found with the lowest levels of cohesion, which coincides with the observed differences in this variable among groups. It has been found that this type of family (disengaged), in which there is little parental–familial closeness, little quality family-time and a lack of common interests, generates little capacity in adolescents to identify and express their emotions, which may explain the feeling of loneliness and exclusion frequently experienced by adolescents who present suicide attempts [35]. 

Regarding adaptability, the most reported style in the group of cases was the structured one (45.0%), which contrast with the percentage of structured families in the control group (23.8%), in which chaotic families were the most frequently found (43.7%); this corresponds to the significant differences in adaptability found among groups, with the chaotic sub-type being the one that shows the highest levels of adaptability. In this sense, although a structured family, which promotes a type of authoritarian leadership, use of democracy, parental hierarchy, establishment of stable functions and an effort to comply with rules, characteristics that can be considered positive in providing stability to the family, they can also condition the presence of high demands and low tolerance of frustration, generating low self-esteem and a feeling of inadequacy [36]. In addition, chaotic families, which are characterized by the presence of undefined roles, absence of leadership and changing or absent discipline [27], may be more functional in the Mexican culture. Nevertheless, more studies are necessary in order to confirm this.

In conclusion, the present study allowed us to describe the characteristics of family functioning in terms of cohesion and adaptability in adolescents with a history of suicide attempts in comparison with a control group, identifying the fact that adolescents with suicide attempts showed lower levels of cohesion and adaptability than cases; according to Olson’s circumplex model, these families are described as structured and disintegrated families; which, although being able to function moderately well in some cultures, may present high self-demand, low tolerance of frustration and little emotional containment generated by the family environment, which is usually associated with the presence of suicide attempts, mainly in the Mexican culture.

The following are recognized as limitations: the size of the sample, the lack of application of the FACES III to any of the parents or family members to corroborate the information, as well as the impossibility of following up to identify changes in the perception of family functioning in both groups. 

More studies, with larger sample sizes and longitudinal designs, which involve all family members and are performed in different cultures, will clarify the role that adaptability and cohesion play in family functioning and suicide attempts in children and adolescents.

## Figures and Tables

**Figure 2 behavsci-13-00120-f002:**
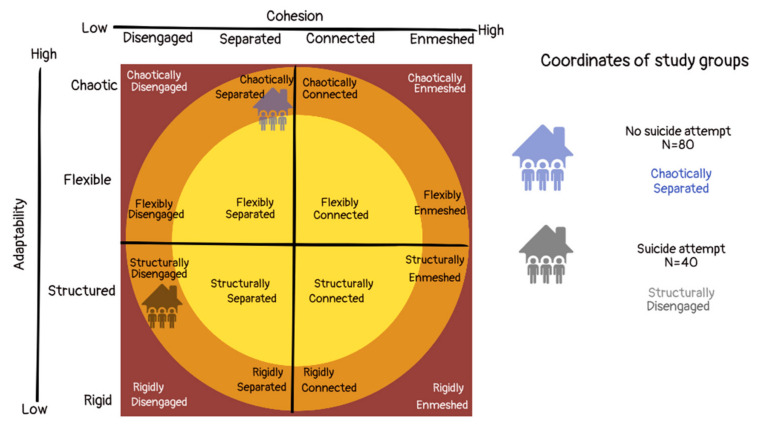
Coordinates of the study groups according to Olson’s circumplex model, showing the family types with the highest frequency found in our study.

**Table 1 behavsci-13-00120-t001:** Comparison of sociodemographic variables between both study groups.

Variable	Cases (N = 40)	Control (N = 80)	Chi-Squared/Z Value/*t*-Test (d.f.)	*p*-Value
Female gender, *n* (%)	32 (80.0)	60 (75.0)	0.373 (1)	0.650
Age, median (range)	15 (12–17)	15 (12–17)	−0.006	0.995
Schooling ^α^, *n* (%)				
Middle-school grade 6	3 (7.5)	4 (5.0)	38.555 (9)	
Middle-school grade 7	4 (10.0)	18 (22.5)		
Middle-school grade 8	13 (32.5)	22 (27.5)		<0.001
High-school grade 9	7 (17.5)	0 (0.0)		
High-school grade 10	8 (20.0)	25 (31.25)		
High-school grade 11	5 (12.5)	11 (13.75)		
Number of siblings, median (range)	2 (0–7)	2 (0–5)	−0.637	0.531
Grade-point average, mean ± SD	8.40 ± 1.01	8.77 ± 0.81	−2.031 (65)	0.046
Maximum Paternal schooling, *n* (%)				
Elementary School	0 (0.0)	3 (3.75)	4.492 (3)	
Middle school	10 (25.0)	31 (38.75)		
High school	18 (45.0)	26 (32.5)		0.236
College	12 (30.0)	19 (23.75)		
Unknown	0 (0.0)	1 (1.25)		
Parental marital status, *n* (%)			7.617 (6)	0.224
Married	17 (42.5)	50 (62.5)		
Divorced	4 (10.0)	7 (8.8)		
Single mother	3 (7.5)	4 (5.0)		
Single father	0 (0.0)	1 (1.2)		
Separated	7 (17.5)	11 (13.8)		
Free union	6 (15.0)	6 (7.5)		
Widow	3 (7.5)	1 (1.2)		
Housing conditions, *n* (%)			5.048 (2)	0.089
Own	25 (62.5)	59 (73.7)		
Rented	8 (20.0)	17 (21.3)		
Borrowed	7 (17.5)	4 (5.0)		
Suicidal attempts, median (range)	2 (1–18)	–	–	–
Psychiatric disorders ^β^, *n* (%)				
Disorder due to use of volatile inhalants (F18.1)	1 (2.5)	–	–	
Single-episode depressive-disorder, severe, without psychotic symptoms (F32.2)	9 (22.5)	–		
Recurrent depressive-disorder, current episode severe, without psychotic symptoms (F33.2)	22 (55.0)	–		
Recurrent depressive-disorder, current episode severe, with psychotic symptoms (F33.3)	2 (5.0)	–		–
Dysthymic disorder (F34.1)	2 (5.0)	–		
Mixed depressive- and anxiety-disorder (F41.2)	1 (2.5)	–		
Borderline pattern (F60.3)	2 (5.0)	–		
Attention-deficit-hyperactivity disorder (F90)	1 (2.5)	–		

^α^ High school in Mexico includes grade 9 to 11, ^β^ ICD-10 diagnoses, *p* values obtained with chi-squared, Mann–Whitney U test and Student’s *t*-test. d.f.: degrees of freedom; Z values (obtained with Mann–Whitney U test), do not have d.f.

**Table 2 behavsci-13-00120-t002:** Comparison of family dimensions “cohesion and adaptability” among the study groups.

Variable	Cases (N = 40)	Controls (N = 80)	*t*-Test Value (d.f.)	Cohen d test/Cohen r test	*p*-Value
Cohesion	2.86 ± 0.87	3.80 ± 0.72	−6.283 (118)	1.217/0.52	<0.001
Adaptability	2.26 ± 0.58	2.72 ± 0.63	−3.852 (118)	0.746/0.35	<0.001

*p* values obtained with Student’s *t*-test. Range of the instrument: 1–5. D.f.: degrees of freedom.

**Table 3 behavsci-13-00120-t003:** Comparison of family sub-types according to adaptability and cohesion in both groups.

Variable	Suicide Attempt N = 40 (%)	No Suicide Attempt N = 80 (%)	Chi-Squared (d.f.) *p*-Value
Cohesion, *n* (%)			
Disengaged	29 (72.5)	22 (27.5)	25.709 (3) <0.001
Separated	9 (22.5)	25 (31.2)	
Connected	2 (5.0)	21 (26.3)	
Enmeshed	0 (0.0)	12 (15.0)	
Adaptability, *n* (%)			
Rigid	10 (25.0)	8 (10.0)	16.038 (3) 0.001
Structured	18 (45.0)	19 (23.8)	
Flexible	7 (17.5)	18 (22.5)	
Chaotic	5 (12.5)	35 (43.7)	

*p* value obtained with chi-squared test, d.f.: degrees of freedom.

**Table 4 behavsci-13-00120-t004:** Comparison of family types.

Variable	Suicide Attempt N = 40	No suicide Attempt N = 80	Chi-Squared (df)	*p*-Value
Family type, *n* (%)				
Extreme	10 (25)	19 (23.8)		0.294
Mid-range	24 (60)	39 (48.7)	2.445 (2)	
Balanced	6 (15)	22 (27.5)		

*p* value obtained with chi-squared test. d.f.: Degrees of freedom.

## Data Availability

Not applicable.

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
