# Peer review of "Family Functioning and Suicide Attempts in Mexican Adolescents"

_behavsci, 2023, doi:10.3390/bs13020120_

Round 1

Reviewer 1 Report

Examining risk factors for suicide attempts in adolescents is highly relevant in many countries. The context is a healthcare crisis and civilizational change. This article has the potential to provide a discussion of risk factors for suicidal behavior from the perspective of family functioning. I suggest the following significant changes that will enhance the value of the article:

1. In the Introduction section, try to clarify the potential importance of the family in creating risk factors for suicide, especially by referring to the concepts introduced (cohesion, adaptability). It is necessary to propose a psychological mechanism (referring to theories of family functioning and knowledge of the theories of suicidality) of how these family characteristics "operate" to create vulnerability. It is helpful to refer to the classic literature, here omitted, which is the work of Salvador Minuchin. It is worth enriching this section. 

2. The rationale for undertaking the research described is present in the Introduction section, but the first two paragraphs of the Discussion section also address this. I suggest moving the first two paragraphs of the Discussion to the Introduction and editing there a coherent justification for the choice of the research problem. Especially since it only became apparent after reading the Discussion that the aim of the study is not only to compare adolescents after suicide attempts with the control group in terms of family characteristics but also in terms of socioeconomic factors and academic performance.

3. It would be necessary to articulate the hypotheses posed in this research. 

4. The phrase "the same procedure was carried..." in the section "Subject recruitment" is misleading, as the procedure for recruiting the clinical group is described above, and the control group was probably not recruited at the hospital after all. More information on the recruitment of the control group is needed. 

5. The last sentence in the section "Subject recruitment" is unnecessary as it states the obvious. 

6. Reliability needs to be added to the description of the FACES III tool.

7. The ICD-10 codes are given in Table 1, but they should be explained in some note or text, as only some know them at a glance. 

8. Punctuation needs to be improved, particularly concerning the presentation of the results of statistical tests. Various labels are used for p-value, but please review punctuation and elsewhere.

9. The captions (titles) above the tables are not complete. They should include a description of the table content, N, and the name of the statistical test. If this information is not in the title itself, it should be readily available in the text. An example is Table 3, where it is unclear what method was used to compare family types. 

10. The division into family types should be theoretically justified, and ideally, this typology should already be introduced in the Introduction section if it is important and necessary. 

11. in Table 4, the p-value is given with a huge number of zeros after the decimal point. I suggest assuming the number of zeros after the decimal point is the same throughout the text. This table also does not show the size of the statistical coefficient used in this comparison. 

12. There is a paragraph on divorce in the Discussion section - I suggest better embedding the explanation of divorce as a risk factor in the literature, as it now sounds intuitive. 

13. An essential point - when talking about differences between groups, more is needed to give the statistical coefficient and significance. What is needed is the effect size. I recommend supplementing all comparisons with this element. This will allow assessing how big the observed statistically significant difference is.

Author Response

Dear reviewer, we thank all your comments and we hope that this manuscript is better for readers now. Please find attached point-by-point answers.

Reviewer 2 Report

Response to authors

Family functioning and suicide attempts in Mexican adolescents

Dear Authors,

I would like to express my gratitude for the opportunity to read this research. The recommendations in this document are intended to help you improve your work. Here are a few small points to consider:

Abstract:

1.    I would suggest authors to add information about subject characteristic and make it clear if the study quantitative study or experimental study because the authors used the control group in method.

2.    In key words, the authors put FACET III, so you should explain why you used FACT III in this study why is not another scale in abstract and in introduction?

Introduction:

1.    In view of the importance of this research, researchers should explain why the authors want to know about family cohesion and adaptability from the point of view of Olson's circumplex model?

2.    Author should explain why used two group in this study.

3.    I propose authors to write the research question and hypotheses of the study before writing the aim of the study. It would be helpful the author to write the results and discussion.

Subject and method:

1.      Please carefully read the sentences ’least 1-day hospital stay was considered as inclusion criteria the same procedure was carried out in a secondary school (High school) where the control group was recruited.

a.        You should explain why 1-day hospital as inclusion criteria?

b.      Why you used the control group is it experimental research?

c.       the same procedure was carried out in a secondary school (High school) where the control group was recruited. You should make it more clear for not make the reader confused with this sentence.

d.      I propose authors to explain the procedure of the research, the treatment for the experiment group to make it clear enough

e.       What the urgency of the subject in the Hospital and in the school? You should put the reason that you learned from your previous investigation.

f.        If it the experimental research, please explain the treatment and how to analysis it.

g.      You should explain Olson's circumplex model in the method...?

Conclusion

The conclusions are adjusted to the introduction and research objectives. Please also check that you have used the references according to these guidelines.

My best regard,

Reviewer 

Author Response

Dear reviewer

We would like to thank all your suggestions and comments; we think that the manuscript has improved significantly. Please find attached the point-by-point answers.

Round 2

Reviewer 1 Report

The article has been significantly improved. I still experience a slight sense of incoherence while reading. This comes also from the fact that although the authors have improved specific pieces of text as suggested, I don't think they have fully assimilated them. For example, in light of the suggestion to report effect sizes, these were reported in the table. However, the significance level was left as a crucial result in the abstract. [I recommend to change this notion in abstract and replace it with the coeficient's value or efect size, that is very high]. At the same time, the topics are important and the results obtained are interesting with interpretive caution.

Author Response

Dear reviewer please find attached our correction
